# Very-Low-Frequency Spike–Wave Complex Partial Motor Seizure Mimicking Canine Idiopathic Head Tremor Syndrome in a Dog

**DOI:** 10.3390/vetsci10070472

**Published:** 2023-07-19

**Authors:** Mihai Musteata, Raluca Ștefănescu, Denis Gabriel Borcea, Gheorghe Solcan

**Affiliations:** 1Neurology Service, Faculty of Veterinary Medicine, Ion Ionescu de la Brad Iași University of Life Sciences, 700489 Iași, Romania; raluca.stef@yahoo.ro (R.Ș.); borceadega@yahoo.com (D.G.B.); 2Internal Medicine Clinic, Faculty of Veterinary Medicine, Ion Ionescu de la Brad Iași University of Life Sciences, 700489 Iași, Romania; gsolcan@uaiasi.ro

**Keywords:** head tremor, partial epilepsy, electroencephalography, spike–wave complex, dog

## Abstract

**Simple Summary:**

Canine idiopathic head tremor syndrome (CIHTS) represents a condition characterized by an episodic, uncontrolled horizontal (“no”), vertical (“yes”) or rotatory movement of the head in dogs. The condition appears to affect young adults, starting when a dog is less active or at rest and stops after applying a stimulus. CIHTS can be an expression of a partial epilepsy. To date, there is a limited number of CIHTS cases in which an electroencephalographic pattern was identified. In this paper, we describe a dog with clinical signs of CIHTS in which a new pattern of interictal epileptic discharges was identified.

**Abstract:**

Canine idiopathic head tremor syndrome (CIHTS) represents a benign condition characterized by episodic, uncontrolled movement of the head. Even though the condition might be an expression of a partial motor epilepsy, to date, there is a limited number of studies describing the electroencephalographic features. In this report, we describe the case of a dog diagnosed with partial motor epilepsy resembling CIHTS symptomatology, a new slow spike–wave complex pattern similar to that of Lennox–Gastaut syndrome in humans identified on electroencephalographic examination. We also studied the efficacy of phenobarbitone therapy over a period of two years.

## 1. Introduction

Canine idiopathic head tremor syndrome (CIHTS) represents a benign condition characterized by an episodic, uncontrolled horizontal (“no”), vertical (“yes”) or rotatory movement of the head [1]. Other terms for this condition include head-bobber or postural repetitive myoclonus [2]. The condition has no know etiology; in some dogs, the condition seems to have an inherited pathogenesis [3] but it can also be a response to an acquired structural brain condition such as a thalamic lesion. A “yes” head tremor also may accompany midline cerebellar lesions. The episodes start when a dog is less active or at rest, suggesting that these may be a manifestation of postural tremors [4,5]. Due to the fact that during the phenomenology the animals appear to be alert, the cessation of the motor activity and the fact that the animal can walk and respond to commands, the postural tremor hypothesis was highlighted [4]. The condition was described in Doberman Pinschers and English Bulldogs [1,3,6], but other breeds can be affected. The reported median age of onset is 3 years and 3 months (range, 3 months to 12 years) [4] and in a half of the patients, the symptoms disappear during their lifetime [6]. The duration of an CIHTS episode lasts from seconds to hours (usually less than 5 min) and their frequency in time is variable [1].

Due to its episodic occurrence, a normal physical and neurological status between the episodes, negative cerebrospinal fluid and advanced imaging examination, a partial epileptic seizure may be considered [4,7] although the efficiency of antiepileptic drugs in treating the condition is questionable. A variety of antiepileptic drugs was advanced as a potential treatment for head tremors. In a group of 43 dogs diagnosed with CIHTS, a therapeutic effect after oral maintenance was seen in 5/20 dogs treated with phenobarbitone and only in 3/12 dogs treated with diazepam [1]. Imepitoin is a low-affinity partial agonist at the benzodiazepine (BDZ) binding site of the GABA_A_ receptor [8] and is commonly used in the chronic treatment of idiopathic epilepsy [9,10], fear and anxiety [8]. In contrast with its success in controlling both the intensity and frequency of epileptic attacks, in dogs with CHITS, imepitoin seems to not offer a significant benefit [11]. In dyskinetic dogs, acetazolamide and fluoxetine appear to have a better clinical response [12]. Furthermore, a gluten-free diet might be considered in dogs suffering from dyskinesia secondary to gluten intolerance [13,14,15].

Identifying an equivocal epileptiform activity during the interictal EEG in dogs with head tremors might be not always successful [2,16]. On the other hand, in some dogs with CIHTS, a new EEG pattern was identified and suggests the reclassification of this medical condition as an epileptic syndrome [17]. To date, the available reports with positive electroencephalographical findings in dogs with CIHTS is limited. Hence, further EEG investigation during and after HT episodes are warranted.

In this paper, we describe a new electroencephalographical pattern that was observed in a dog with a clinical symptomatology suggestive for CIHTS.

## 2. Case Description

A 1.5-year-old crossbred, 7.4 kg, Bichon Frise mix intact male dog, with a three-month history of recurrent, involuntary episodes of head tremor was referred to our neurology service for further investigation. The frequency of the episodes increased from two episodes per week to multiple episodes in a couple of hours. Each episode lasted from 3 min to up to 5 h. The owner described the majority of the episodes as having occurred when the dog was lying down in a quiet but awake state and were characterized mainly by a vertical (“yes”) movement of the head with only a few episodes with horizontal (“no”) head movements. The motor activity episode cessation was induced when the dog’s attention was distracted, but in some cases, a new episode started after several seconds (see Appendix A). The owner reported that, a week prior the admission in our hospital, the dog elicited a long episode of head tremor (>1 h) and intrarectally diazepam (0.5 mg/kg) was administered by the veterinarian. After diazepam administration, the head tremor stopped but a new episode occurred in less than 30 min. With the exception of these episodes, no other abnormalities were reported in his medical and physical records.

At admission, no abnormalities were observed during general examination (38.6 °C, heart rate 120 beats/min, respiratory rate 30/min with a capillary refill time inferior to 1 s). The dog had a normal mentation and no abnormalities were observed on proprioceptive tests or when spinal reflectivity and cranial nerves were examined. The palpation of vertebral column and neck manipulation was not painful.

No abnormalities were noticed on cell blood count and serum biochemistry (total calcium 10.5 mg/dL, creatine phosphokinase 380 U/L, total protein 6.5 g/dL, alkaline phosphatase 70 U/L, aspartate aminotransferase 30 U/L, alanine aminotransferase 73 U/L, creatinine 0.81 mg/dL, and urea 45.3 mg/dL). Thoracic X-ray (Maxivet 400 HF DR, Comes Electro, Cavaria Con Premezzo, Italy), abdominal ultrasound (LogiqV5 ultrasound machine with a 4–8-MHz phased-array probe, General Electric Medical Systems, Wuxi, China) were in the normal limit and a 5 min six-lead ECG examination (ECG; PolySpectrum 8 V ECG equipment, Neurosoft, Ivanovo, Russia) revealed a normal ECG trace morphology. The blood pressure was assessed according to a previously described methodology [18] and the patient was considered normotensive, with a systolic blood pressure (mean (m) ± standard deviation (SD)) of 123  ±  9 mmHg and diastolic blood pressure (m  ±  SD) of 73  ±  2 mmHg.

A cerebrospinal tap, brain magnetic resonance (MRI) study and EEG examination was proposed. The owner declined the MRI examination.

The CSF sampling was taken by puncture at the atlantooccipital space, under general anesthesia with xylazine (Xylazin Bio, Bioveta a.s., Komenského, Czech Republic) 1 mg/bw and ketamine (Ketamine, Kepro, Deventer, the Netherlands) 0.1 mg/bw inj. IV. The CSF was considered normal in terms of the number of cells (2 cells/mL, reference value <5 cells/mL [19]) and protein level (10 mg/L, reference value <300 mg/L [19]).

The electrophysiological tests were performed after two days under sedation with xylazine (Xylazin Bio, Bioveta a.s., Komenského, Czech Republic) 1 mg/bw using a Neurofax electroencephalograph (Nihon Kohden, Tokyo, Japan). After sedation, biopotentials were acquired for a minimum of 30 min with 0.22 × 22.5 mm and 1.5 mm proof connector subdermal stainless-steel electrodes (NE-224S EEG, Nihon Kohden, Tokyo, Japan). A short time EEG (30 min) with a 7-channel examination was recorded. The electrodes were placed according to previously described protocols [20,21] as follows: F3, F4 C3, C4, Cz, O1 and O2 for a monopolar montage with reference electrodes (A1, A2) placed on the corresponding ear. For a bipolar montage, the reference electrode was placed on the nasal bone. The electroencephalographic recording was conducted using the parameters: sensitivity 75 mV, time constant 0.3 s, filter pass-down 70 Hz, filter pass-up 30 Hz, and electrode impedance <10 Ω.

During the sedation, the EEG trace was characterized by a normal low-voltage, high-frequency (freq. > 8 Hz, amplitude 10–20) background activity (Figure 1) [20]. Amid the recovery, once the animal was able to sustain its head and to be alert and responsive to the environment, short episodes (<20 s) of a rhythmic activity represented by spike–wave complexes (SWCs) were observed in both monopolar and bipolar montages of the EEG traces. The complexes were characterized by a very-low frequency (1.5–2 Hz) and large amplitude (up to 400 µV). Some of the SWC episodes, visible on EEG, coincided with the presence of “yes-yes” movements of the head similar to those described in the anamnesis.

Based on the clinical, paraclinical and electrodiagnostic data, a presumptive diagnosis of partial motor epilepsy was established and a 3 mg/kg/12 h p.o. phenobarbitone (Fenobarbital, Zentiva s.a., Bucharest. Romania) therapy was initiated. After two weeks, a normal phenobarbitalemia was documented (29.1 µg/mL; normal range 20–35 µg/mL) and the initial dosage was maintained. Over the next year, the frequency of head tremor episodes diminished to one episode at every 4 months and the duration was shorter to 3 min. After a year, the owner voluntarily diminished the phenobarbital dosage (3 mg/kg/24 h) and three episodes in one single week were documented. At the moment of reexamination, the phenobarbitalemia was determined to be out of therapeutic ranges (<5 µg/mL). The owner was instructed to maintain the prescribed dosage and during the next 13 months the dog had only two episodes which last less than 2 min each.

## 3. Discussions

In this paper, we described a dog with a presumed partial motor seizure clinically mimicking CIHTS. The electroencephalographic pattern was represented by high-amplitude 1.5–2 Hz spike–wave complexes clinically corresponding to a “yes” movement of the head.

Head tremors may be observed in different structural conditions of the central nervous system (brain and cervical spinal cord) in dogs [22,23], cats and humans [24]. However, in all of those situations, other neurological deficits are usually present between the episodes of head tremor. Moreover, certain exceptional conditions such as illness, surgery, medication, heat, pseudopregnancy or pregnancy have been reported to trigger head tremor episodes. CHITS is a nonprogressive disorder; hence, the quality of life of affected dogs is usually not affected, even if a long episode [25] may be perceived as a stressful for both the animal and owner. By its episodic occurrence, CHITS exhibits similarities with epileptic and dyskinetic events; in consequence, multiple antiepileptic drugs were used to try to decrease the frequency and intensity of the episodes. Unfortunately, the reported therapeutic efficacy of antiepileptic drugs seems to be an unsatisfactory one.

EEG examination is a recommendation of the International Task Force in Veterinary Epilepsy in the diagnostic protocol of a patient with presumed idiopathic epilepsy [16]. However, the lack of standardization or the availability of the equipment makes the technique not commonly used in the diagnosis of epileptic, dyskinetic or other brain disorders. In a study performed in ten dogs with CIHTS, a right-sided, predominant, 6 Hz, spike-and-wave discharges associated with the head tremor was observed during the ictus in 3 of 10 dogs and a 12–14 Hz polyspike-and-wave pattern in another one. Based on those findings, the authors suggest the reclassification of CIHTS as an epileptic syndrome of 6 Hz spike-and-wave patterns with focal onset seizures in some dogs [5].

In our dog, EEG epileptic discharges were characterized by a complex of spike–wave discharges with a high amplitude and very low frequency (<2 Hz). Low-frequency spike–waves discharges were previously reported both in human and veterinary medicine. A 4 Hz SWC discharge, indicative of epilepsy with myoclonic absences, was reported in absence seizures in a Chihuahua dog [26] correlated with episodes of intermittent hind limb jerks and head twitching along with nose twitching and features appearing somehow “disconnected” and transiently impaired in his behavioral attitude. The authors stated that the identified dog’s 4 Hz SWC pattern resembled the 3–4.5 Hz SWC pattern seen in human with typical epileptic absences. On the other hand, atypical absence seizures, with a lower frequency SWC (<2.5 Hz) are observed in human patients in symptomatic epilepsies and impaired cognition conditions (e.g., Lennox–Gastaut syndrome). The characteristic EEG abnormalities in Lennox–Gastaut syndrome are represented by the bursts of diffuse slow spike–waves at 2–2.5 cycles/s in the interictal EEG when the patient is awake or bursts of fast rhythmic waves and slow polyspikes and generalized fast rhythms at about 10 cycles/s during sleep [27]. No cognitive impairment was identified in our case during the examinations or reported by the owner and no other motor components were observed during the head tremor episodes. Yet, it is disputable if the tongue movements observed both in the videos provided by the owner (Appendix A) and during the EEG recording episodes are equivalent with motor epileptic automatisms.

In our dog, the phenobarbitone therapy was effective and offered a good control of the episodes as long as the owner followed the recommended posology. Interestingly, in the second year after the initial presentation, the frequency of the head tremor episodes decreased dramatically (no documented epileptic episodes for more than seven months). CIHTS is frequently reported in young dogs (around 3 years old) and the condition appears to be self-cured in time in a large number of dogs. In this light, our dog might have the same clinical course and no antiepileptic drug will be needed in time. In such a scenario, CIHTS with epileptic component (low frequency SWC on EEG) should be considered more than partial motor epilepsy mimicking CIHTS.

The limitation of our study is represented by the absence of an MRI study to exclude a potential brain/cervical spinal cord structural etiology (tumors, inflammation, anomalies, etc.). However, any of those hypotheses would have evolved in time and additional permanent neurological deficits would be expected to be seen between episodes [12]. Hence, a structural intracranial or spinal condition should be not privileged. In the presented dog, after more than two years of follow ups, no neurological deficits were identified and no drug able to mask/suppress the symptomatology other than the phenobarbitone was administered, sustaining the initial diagnosis.

## 4. Conclusions

Here, we described a dog with a very-low-frequency spike–wave electroencephalographic pattern motor epilepsy mimicking CIHTS. Our data contribute to the limited number of reports documenting idiopathic head tremor syndromes in dogs and describe a new EEG feature similar to that of Lennox–Gastaut syndrome in humans. The phenobarbitone therapy offered an effective clinical response.

## Figures and Tables

**Figure 1 vetsci-10-00472-f001:**
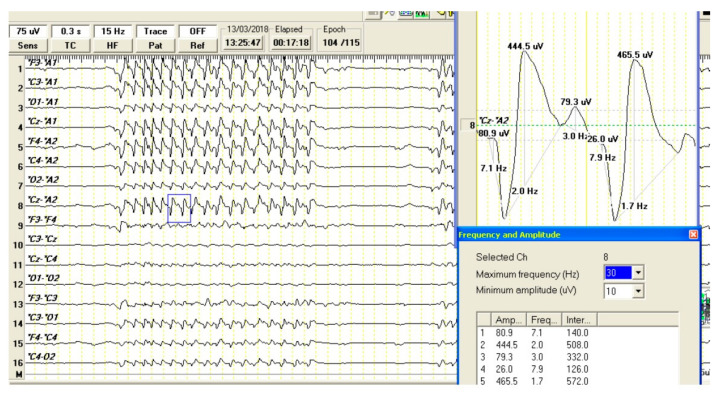
Electroencephalogram of a dog with clinical symptoms resembling canine idiopathic head tremor syndrome during the recovery from xylazine sedation. Monopolar (ears as reference A1, A2; channels 1 to 8) and bipolar montages (F3-F4, C3-Cz, Cz-C4, O1-O2, F3-C3, C3-O1, F4-C4, C4-O2; channels 8 to 16) are presented. The recording was made at a sensitivity of 75 mV, time constant of 0.3 s, filter pass-down of70 Hz, filter pass-up of 30 Hz, and electrode impedance <10 Ω. A short episode (<20 s) of a rhythmic activity represented by spike–wave complexes (SWCs) disrupted the normal low-voltage, high-frequency (freq. > 8 Hz, amplitude 10–20) background activity. Two selected SWCs (blue box) characterized by a very low frequency (1.7 and 2 Hz) and high amplitude (444.5 and 465.5 µV, respectively) are presented in detail.

## Data Availability

The article includes all relevant study data.

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
