# Peer review of "Very-Low-Frequency Spike–Wave Complex Partial Motor Seizure Mimicking Canine Idiopathic Head Tremor Syndrome in a Dog"

_vetsci, 2023, doi:10.3390/vetsci10070472_

Round 1

Reviewer 1 Report

The manuscript by Musteata et al. is a case report about a Bichon Frise mix intact male dog, with clinical signs of canine idiopathic head tremor syndrome and with a new pattern of interictal epileptic discharges.

A partial motor epilepsy diagnosis is taken following clinical, paraclinical and electrodiagnostic data (EEG). The owner declined the MRI, so it is not possible to 100% ensure an idiopathic cause and to exclude other causes (i.e. malformation or neoplasia).

A phenobarbitone therapy is here highlighted and it could be the therapy of choice to control the head tremor episodes.

The case is well presented.

The discussion is well structured, interesting and easily readable and encompasses all the main key points of this case.

Some moderate english editing is necessary.

As an exemple, in line 60: “Each episode last from 3 minutes to up to 5 hours.” Lines 172-173: In a such a scenario an CIHTS with epileptic component 

Etc.

Author Response

We kindly appreciate your time and effort in reviewing our work. Thank you for your suggestions. We believe that in the revised form the language mistakes were solved.

Reviewer 2 Report

General comments:

This manuscript is about partial motor seizure mimicking idiopathic head tremor syndrome, which is a very interesting theme with clinical significance. However, there are some limitations, such as the fact that there is only one case and it should have more documentation. I think this manuscript has potential for publication with minor revisions. 

Specific comments:

Introduction section (paragraph 28 - 42): I suggest the authors add more bibliographic references in this paragraph.

Case description:

- Line 71:  When the authors talk about the physical examination and the neurological examination, I suggest exploring these and giving a detailed neurological examination. 

- Line 72: What biochemistry analyses were done?

- Line 72: What was the x-ray device? Authors should describe as they did with ultrasound machine. 

- Line 79: There is a dot missing in the final.

Also, I think it would be very interesting for the readers to add some pictures, for example, of the electrode placement.

Author Response

We kindly appreciate your time and effort in reviewing our work. Thank you for your suggestions. Please find bellow our answer to your comments:

C1: Introduction section (paragraph 28 - 42): I suggest the authors add more bibliographic references in this paragraph.

A1: In the revised form, the introduction section is more detailed and, in consequence, new references were added.

C2: Line 71: When the authors talk about the physical examination and the neurological examination, I suggest exploring these and giving a detailed neurological examination

A2: Thank you for your suggestion. In the revised version details about the neurological examination were added.

C3: What was the x-ray device? Authors should describe as they did with ultrasound machine.  and Line 72: What biochemistry analyses were done?

A3: We included the missing information.

C4: I think it would be very interesting for the readers to add some pictures, for example, of the electrode placement.

A4: Thank you for your remark. Indeed, we agree that by adding a picture with the electrode placement might be useful for readers who are less familiar with the EEG technique. Unfortunately, we have no quality photo with this case able to satisfy the journal standards. Hence, in order to compensate that, immediately after we described the EEG protocol we included relevant references where helpful graphical representation of the electrode placement is presented (Wrzosek, M., 2017 and Pellegrino, F.C.; 2004)  

Reviewer 3 Report

Manuscript ID: vetsci-2457495

Title: Very low frequency spike-wave complex partial motor seizure mimicking canine idiopathic head tremor syndrome in a dog

In the manuscript authors represent a dog with very low frequency spike-wave electroencephalo- graphic pattern motor epilepsy which mimicking canine idiopathic head tremor syndrome (CIHTS).

Authors can describe more about Canine idiopathic head tremor syndrome (CIHTS) in dogs in the introduction section, such as disease occurrence cause, initial study and present scenario of disease and current treatment or medicines which are being used at present.

Authors present only Electroencephalogram data but other data such as Biochemical blood test results and X-Ray data should be mentioned in the manuscript.

Authors mentioned old study reference (before 2016) in the manuscript. Authors should mention new research data references in whole manuscript.

Line 40-42: Reference is missing

Line 56-57: The sentence was not clear, what that intend by mentioning “A 1,5 -year-old crossbred, 7,4 kg, Bichon Frise mix intact male dog, was admitted in the Neurology Service of VTH Iași, for evaluation of recurrent episodes of head tremor”.

Line 100-104: Sentence is too long and unclear, should be rewritten.

Figure 1: Figure quality is not up to mark for publication, enhance figure quality (Must be TIF file) and must be revised figure legend, it should be self-sufficient to explain.  

Line 147: In place of patient should write in bracket “Dog”

“In our patient EEG epileptic discharges”

Author Response

We kindly appreciate your time and effort in reviewing our work. Thank you for your suggestions. Please find bellow our answer to your comments:

C1: Authors can describe more about Canine idiopathic head tremor syndrome (CIHTS) in dogs in the introduction section, such as disease occurrence cause, initial study and present scenario of disease and current treatment or medicines which are being used at present.

A1: Thank you for your suggestion. We completed the Introduction section accordingly.

C2: Authors present only Electroencephalogram data but other data such as Biochemical blood test results and X-Ray data should be mentioned in the manuscript.

A2: We included the required data.

C3: Authors mentioned old study reference (before 2016) in the manuscript. Authors should mention new research data references in whole manuscript.

A3: We appreciate your remarks. The number of papers about CIHTS is limited. However, in the revised form we detailed more the Introduction section. Hence, we included references from the 2016-2023 too. We hope that in the actual form, the manuscript responded in full to  the abovementioned remarks.

C4: Line 40-42: Reference is missing

A4: In the revised form we included the reference.

C5: Line 56-57: The sentence was not clear, what that intend by mentioning “A 1,5 -year-old crossbred, 7,4 kg, Bichon Frise mix intact male dog, was admitted in the Neurology Service of VTH Iași, for evaluation of recurrent episodes of head tremor”.

Line 100-104: Sentence is too long and unclear, should be rewritten.

A5: We appreciate your suggestion. In the revised form we rephrased the paragraphs.

C6:  Figure 1: Figure quality is not up to mark for publication, enhance figure quality (Must be TIF file) and must be revised figure legend, it should be self-sufficient to explain.  

A6: We appreciate your comment. In the revised form, we change the figure file with a TIF file one. Accordingly, we revised the legend and we hope that in this form we responded to this issue.  

C7: Line 147: In place of patient should write in bracket “Dog” “In our patient EEG epileptic discharges”

A7: Thank you for your comment. We made the change accordingly.

Once more, the authors thank to the reviewer for his valuable comments and suggestions. We found those helpful in improving the quality of the manuscript!

Reviewer 4 Report

the paper is well written and excellent described

Author Response

Thank you for your time and effort in reviewing our manuscript! We really appreciate your feedback!